# Anterior Gradient Protein 3 and S100 Calcium-Binding Protein P Levels in Different Endometrial Epithelial Compartments May Play an Important Role in Recurrent Pregnancy Failure

**DOI:** 10.3390/ijms22083835

**Published:** 2021-04-07

**Authors:** Nicola Tempest, Elizabeth Batchelor, Christopher J. Hill, Hannan Al-Lamee, Josephine Drury, Andrew J. Drakeley, Dharani K. Hapangama

**Affiliations:** 1Centre for Women’s Health Research, Department of Women’s and Children’s Health, Institute of Life Course and Medical Sciences, University of Liverpool, Member of Liverpool Health Partners, Liverpool L8 7SS, UK; elizabatchelor@hotmail.co.uk (E.B.); c.j.hill1@liverpool.ac.uk (C.J.H.); allameeh@gmail.com (H.A.-L.); jadrury@liverpool.ac.uk (J.D.); dharani@liv.ac.uk (D.K.H.); 2Liverpool Women’s NHS Foundation Trust, Member of Liverpool Health Partners, Liverpool L8 7SS, UK; 3Hewitt Centre for Reproductive Medicine, Liverpool Women’s NHS Foundation Trust, Liverpool L8 7SS, UK; Andrew.Drakeley@lwh.nhs.uk

**Keywords:** AGR3, S100P, fertility, reproduction, window of implantation

## Abstract

Recurrent implantation failure (RIF) and recurrent pregnancy loss (RPL) are distressing conditions without effective treatments. The luminal epithelium (LE) is integral in determining receptivity of the endometrium, whereas functionalis glands and stroma aid in nurturing early embryo development. Calcium signalling pathways are known to be of vital importance to embryo implantation and pregnancy establishment, and anterior gradient protein 3 (AGR3) and S100 calcium-binding protein P (S100P) are involved with these pathways. We initially examined 20 full-thickness endometrial biopsies from premenopausal women across the menstrual cycle to characterize levels of AGR3 protein in each endometrial sub-region at the cellular level. A further 53 endometrial pipelle biopsies collected in the window of implantation were subsequently assessed to determine differential endometrial AGR3 and S100P levels relevant to RIF (*n* = 13) and RPL (*n* = 10) in comparison with parous women (*n* = 30) using immunohistochemistry. Significantly higher AGR3 and S100P immunostaining was observed in ciliated cells of the LE of women with recurrent reproductive failure compared with parous women, suggesting aberrant subcellular location-associated pathophysiology for these conditions. The nuclear localisation of S100P may allow transcriptional regulatory function, which is necessary for implantation of a viable pregnancy. Further work is thus warranted to assess their utility as diagnostic/therapeutic targets.

## 1. Introduction

Recurrent reproductive failure is a devastating reproductive health issue that may manifest in two ways: (i) recurrent implantation failure (RIF) during in vitro fertilization (IVF) treatment cycles or (ii) recurrent pregnancy loss (RPL). RIF affects approximately 10% of women undergoing IVF [1,2], and the broad definition incorporates the number of previous IVF embryo transfer (ET) failures as a diagnostic criterion, with three failed IVF-ET attempts being the most widely accepted threshold [3,4]. There are no universally accepted definitions of RPL, and the loss of two [5,6] or three [7] consecutive pregnancies [6,7] have been proposed, with most studies reporting the prevalence to be 1–2% of women [5,7]. Unexplained RPL and RIF have no effective treatments, and presumably the causes are multifactorial, although a hostile endometrial environment is likely implicated. Both conditions are highly distressing for patients and treatment constitutes one of the most difficult challenges for clinicians in the field [3,8,9].

The luminal epithelium (LE) (the uppermost layer of the endometrium) is the first maternal layer of cells that an embryo communicates with during the window of implantation (WOI) [10]; therefore, it is central to determining the receptivity of the endometrium [11]. The LE is a single layer of cuboidal epithelial cells, less abundant than their glandular counterparts, and therefore could be underrepresented in studies that use either whole endometrial tissue or isolated epithelial cells [11]. In the follicular phase of the cycle, the LE possess many ciliated cells [12] located near gland orifices at tubal ostia and close to the endo-cervical mucosa [13]. The role of the cilia is to remove the secretions of the bordering cells, and aid in spermatozoa kinetics and in the capitation of the oocyte. The LE undergoes apical surface specialisation, expressing cell adhesion molecules that permit adherence of the blastocyst [14] and controls excessive trophoblastic invasion [15]. The functionalis glands undergo massive proliferation in response to oestrogen in the proliferative phase, and differentiate under progesterone regulation in the secretory phase. Their secretions play a fundamental role during early pregnancy establishment by secreting substances that support blastocyst development [14], confirmed by its absence being associated with reduced survival of the conceptus [10].

Calcium signalling pathways are of vital importance to embryo implantation and early pregnancy establishment [16,17]. Anterior gradient protein 3 (AGR3) is a protein that has recently been shown to be restricted to ciliated cells in the airway epithelium, where it plays an important, calcium-mediated role in the regulation of muco-ciliary function in the airway. AGR3 is required for regulation of ciliary beat frequency and may also be involved in the regulation of intracellular calcium in tracheal epithelial cells [18]. Endometrial expression and the functional relevance of AGR3 is unknown but AGR2, which shows a 71% sequence homology with AGR3, was previously shown to be hormonally regulated in the endometrium [19]. AGR3 protein was first identified in breast cancer cell lines [19]. Since then, it has been found to be highly expressed in many malignant tissue types and upregulated by both androgens and oestrogens in prostate cancer cell lines [20]. Moreover, research involving both the benign and malignant cells of the prostate showed AGR3 expression to be affected by androgens. Additionally, AGR3 was proposed as a novel serum-based biomarker for breast cancer [21].

S100 calcium-binding protein P (S100P) was originally discovered in 1992 from placental isolates and is a member of the family of S100 low molecular weight calcium binding proteins implicated in calcium sensing and signal transduction [22,23]. Most previous work on S100P has been in the field of cancers, where it has been shown to play vital roles in cellular proliferation, invasion, survival, and angiogenesis [24], all of which are fundamental functions in embryo–endometrial implantation. S100P has previously been shown to be involved in endometriosis and endometrial cancer, and demonstrates highest endometrial levels during the WOI [23,25]; thus, it is proposed to play a vital role in embryo implantation and is a possible candidate marker for endometrial receptivity.

In RIF, good-quality embryos fail to attach to the endometrium, potentially highlighting a defect in the LE, whereas in RPL, pregnancies fail to progress after implantation; thus, endometrial functionalis glands and stroma are anticipated to play an important role in the pathophysiology. Examining particular proteins important to these sub-endometrial areas may provide novel diagnostic and therapeutic targets. Considering the important role of AGR3 in the regulation of ciliary beat frequency and high S100P expression in the WOI, we hypothesised that women with recurrent reproductive failures (RIF or RPL) could have altered levels of these calcium-binding proteins in the WOI when compared with parous women, thus playing an important role in embryo transmission and implantation.

## 2. Results

AGR3 levels in healthy human endometrium across the menstrual cycle have not been previously demonstrated, and initially we assessed the immunoexpression of AGR3 in healthy, full-thickness premenopausal endometrium from a cohort of 20 women undergoing hysterectomy for non-endometrial pathology (cohort 1, Table 1). We found AGR3 levels to be consistently high in the LE throughout the proliferative and secretory phases of the menstrual cycle compared with the functionalis and basalis epithelium, which demonstrated much lower levels of AGR3 (Figure 1A,B). Using immunofluorescence and the ciliated cell marker acetylated-α tubulin co-localisation with AGR3, we demonstrated similar endometrial AGR3 immunostaining in ciliated epithelial cells as previously observed in the lung (Figure 1C). S100P expression throughout the cycle in cohort 1 was consistent with previous publications [25].

Our test cohort (cohort 2) included 53 patients (10 RPL, 13 RIF, and 30 parous controls) who underwent a pipelle biopsy in the WOI to study both the LE and functionalis layer of the endometrium. The average age of the women included in cohort 2 was 39 years (Table 2); the parous control group had a median of three live births, the women with RPL had a median of three previous pregnancy losses and the women with RIF had a median of eight previous failed ETs.

Higher levels of nuclear, perinuclear (*p* < 0.01), cilial (*p* < 0.05), and cytoplasmic (*p* < 0.01) AGR3 staining scores were seen in the LE compared with the glandular epithelium across all groups (Figure 2A–E). The highest perinuclear AGR3 was observed in the LE of the RIF group compared with the fertile control group (*p* < 0.05) (Figure 2C). Ciliated LE cells in both recurrent pregnancy failure groups (RPL/RIF) demonstrated significantly higher immune scores (*p* < 0.01) compared with the fertile controls (Figure 2D).

S100P immunoreactivity in the epithelial sub-types and the subcellular locations in the healthy fertile endometrium were different to the S100P staining patterns in the endometrial samples harvested from women with RPL and RIF. Nuclear S100P in particular was seen in the LE of the fertile control samples harvested in the WOI, whereas an absence of nuclear S100P immunoreactivity was observed in the WOI of the RPL and RIF samples (Figure 3A,B). Ciliated cells of the LE of the RPL/RIF samples demonstrated significantly higher (*p* < 0.01) S100P immunostaining compared with the control samples (Figure 3A,D).

By examining the Ingenuity Knowledge Base, we were able to identify a mechanistic network between S100P, AGR3, AGR2, and embryo implantation (Figure 4). This network shows direct protein–protein interactions between S100P and AGR2, and AGR3 and oestrogen receptor β (ESR2). Here, we demonstrate a causal relationship between embryo implantation via E-cadherin (CDH1) and progesterone. Progesterone was seen to indirectly lead to expression of S100P.

## 3. Discussion

We described the presence of AGR3 protein in human endometrial epithelial cells for the first time, and demonstrated differential endometrial AGR3 and S100P immuno-expression levels during the WOI in women with a history of RPL and/or RIF. The observed higher immunoreactivity for these two calcium-binding proteins in the ciliated LE cells of women with a history of recurrent reproductive failure suggests their erroneous subcellular location may contribute to the pathophysiology of these conditions. The nuclear localisation of S100P is essential for its transcriptional regulatory function, which may be necessary for implantation of a viable pregnancy (Figure 5).

This study confirms the findings of other previous studies demonstrating that different human endometrial sub-regions are distinctive in terms of their expression profiles. Previous microarrays of laser capture micro-dissected LE and glandular epithelium taken at luteinising hormone (LH)+2 and LH+7 revealed distinct mRNA profiles that correlated with immunohistochemistry-based assessment of the corresponding protein expression levels [11]. Despite this knowledge, microarray-based commercial tests for endometrial receptivity assess the broken-down whole endometrial biopsy as a single entity. This approach is less precise due to dilution or masking of the region- or cell-type-specific aberrations relevant to a particular pathology.

Although we only employed immunohistochemistry, and no functional studies have been conducted to confirm a role for the two calcium binding proteins we investigated in the implantation process, significant differences were identified and the importance of tissue positional integrity was highlighted. Further work is warranted to assess their utility as diagnostic and therapeutic targets. Although our initial hypothesis also included that different staining patterns would be seen between women with RPL in comparison with those with RIF, our data did not demonstrate this difference in the levels of the two proteins we examined in our cohort of women. We excogitate that further discriminatory differences will be present with increased sample size and number of molecules examined, highlighting the need for further work.

The obvious differences seen between women with and without RPL or RIF highlight that both AGR3 and S100P proteins may have a functional role in fertility. This study paves the way for further targeted research regarding the LE and implantation. The endometrium cannot be treated as one entity, and care needs to be taken to ensure the individual elements involved in implantation are studied as such. Future studies require the adoption of appropriate methodologies that appreciate and highlight the cell type and region-specific differences relevant to the endometrial pathology. Only then can progress be made to demonstrate how these differences could be targeted with efficient treatment to positively influence the outcomes of millions of women suffering with recurrent reproductive failure.

## 4. Materials and Methods

### 4.1. Tissue Samples

Full-thickness endometrial biopsies were collected from 20 women (10 in the proliferative phase and 10 in the secretory phase of the endometrial cycle), forming cohort 1 (the cycle phase of the endometrium was assigned according to the last menstrual period (LMP) and histological criteria [26,27]), and pipelle endometrial biopsies were collected in the WOI (urinary LH + 7 ± 2) from 53 women (10 recurrent pregnancy loss (RPL), 13 recurrent implantation failure (RIF) and 30 controls) forming cohort 2, all with no known endometrial pathology or use of hormonal medications for 3 months prior to sampling at Liverpool Women’s NHS Foundation Trust. Collection and use of all samples were approved by the Liverpool Adult Ethics committee (REC references; 19/WA/0271 and 19/SC/0449). Informed written consent was obtained from all participants.

### 4.2. Immunohistochemistry

Immunohistochemistry (IHC) was performed as previously described [28]. Sections of formalin-fixed paraffin-embedded (FFPE) tissues were cut at 3 μm. Dried sections were dewaxed and rehydrated prior to heat-induced epitope retrieval in a pressure cooker containing 0.01 M citrate at pH 6.0 for 2 min. Endogenous peroxidase activity was blocked with 0.3% hydrogen peroxide (Thermo Fisher Scientific, Runcorn, UK) for 10 min, prior to incubation with diluted primary antibody overnight at 4 °C (rabbit anti-human AGR3 polyclonal antibody, HPA053942 diluted 1:200 (Sigma Aldrich, Gillingham, UK); and mouse anti-human S100P (clone 16/S100P), diluted 1:100 (BD Biosciences, Wokingham, UK). The appropriate ImmPRESS polymer-based system was applied for 30 min at room temperature, and visualisation was with ImmPACT DAB, following the manufacturer’s instructions (Vector Laboratories, Peterborough, UK). Sections were counterstained using Gill II Haematoxylin (Thermo Fisher Scientific), dehydrated, cleared, and mounted using Consul-Mount (Thermo Fisher Scientific). Matching isotype replaced primary antibody as a negative control, with an internal positive control in each staining run. Slides were digitalised using an Aperio CS2 slide scanner (Leica Biosystems, Milton Keynes, UK).

### 4.3. Immunofluorescence

Immunofluorescence staining was performed on 3 μm FFPE tissue sections. Following antigen retrieval in a pressure cooker for 2 min in 0.01 M citrate at pH 6.0, sections were blocked and incubated with primary antibodies (AGR3 1:100, acetylated-α tubulin 1:500 (6–11B-1 Cell Signalling Technology, London, UK)) overnight at 4 °C. Appropriate secondary antibodies were incubated for 2 h at room temperature, and sections were mounted in VECTASHIELD^®^ with DAPI (Vector Laboratories). Images were captured from a Nikon Eclipse 50i microscope using NIS Elements-F software (Nikon Corporation, Tokyo, Japan).

### 4.4. Image Analysis

Percentage positive (%P) and quick score (QS) were used as previously described [28,29] to score subcellular AGR3 immunoexpression in luminal and glandular epithelial cells: nuclear (QS), perinuclear (%P), ciliated (%P), and cytoplasmic (%P) cells. For subcellular S100P immunoexpression in luminal and glandular epithelial cells, nuclear (QS), cytoplasmic (QS), and ciliated (%P) were used. For S100P expression in stromal cells, cytoplasm QS and nuclear QS were used.

### 4.5. Statistical Analysis

Statistical analysis was performed using GraphPad-Prism Version 5. Non-parametric Mann–Whitney U tests were used to compare groups. Statistical significance was inferred when *p* < 0.05 (*), highly significant when *p* < 0.01 (**), and very highly significant when *p* < 0.001 (***). Box and whisker plots show the median (horizontal line), interquartile range (box), and data range (whiskers).

### 4.6. Ingenuity Pathway Analysis

A mechanistic network was created using the Ingenuity Knowledge Base via the Ingenuity Pathways Analysis software (IPA, Ingenuity Systems, http://www.ingenuity.com, accessed on 1 February 2021). Path explorer was used to identify specific known molecular interactions between implantation of embryo, S100P, AGR3, and AGR2.

## Figures and Tables

**Figure 1 ijms-22-03835-f001:**
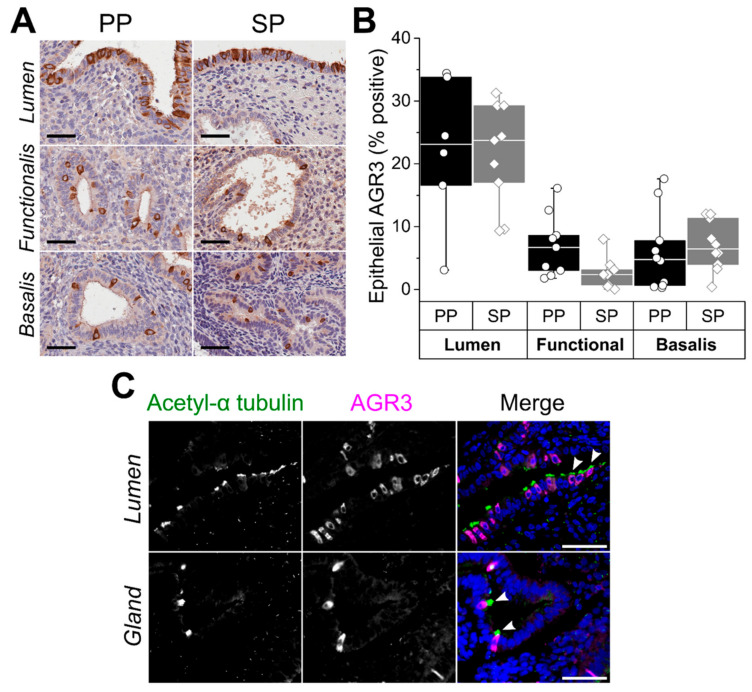
Anterior gradient protein 3 (AGR3) expression in the healthy human endometrium. (**A**) Representative micrographs demonstrate AGR3 immunostaining in proliferative phase (PP) and secretory phase (SP) endometrium. (**B**) Epithelial AGR3 immunostaining scores across subanatomical regions of the cycling endometrium. (**C**) Representative immunofluorescence micrographs show co-staining of acetylated-α tubulin (green) and AGR3 (magenta) in ciliated luminal and glandular endometrial epithelial cells. Tissue sections were counterstained with DAPI (blue). Ciliated cells expressing AGR3 are highlighted with white arrowheads. Scale bars 50 μm throughout.

**Figure 2 ijms-22-03835-f002:**
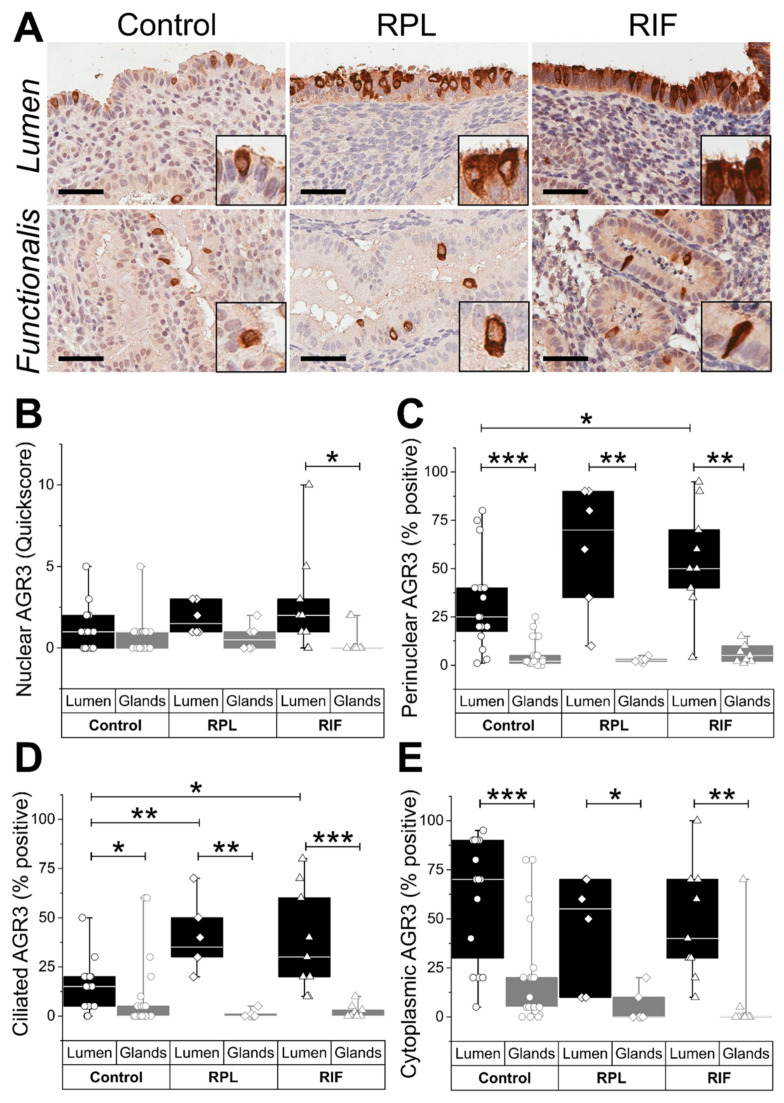
Endometrial AGR3 expression in control and recurrent reproductive failure. (**A**) Representative micrographs demonstrate AGR3 immunostaining in endometrial tissue from fertile (control), recurrent pregnancy loss (RPL), and recurrent implantation failure (RIF) patients. Scale bar 50 μm. Immunostaining scores for (**B**) nuclear, (**C**) perinuclear, (**D**) ciliated, and (**E**) cytoplasmic AGR3 expression in the luminal and glandular epithelium. * *p* < 0.05, ** *p* < 0.01, and *** *p* < 0.001.

**Figure 3 ijms-22-03835-f003:**
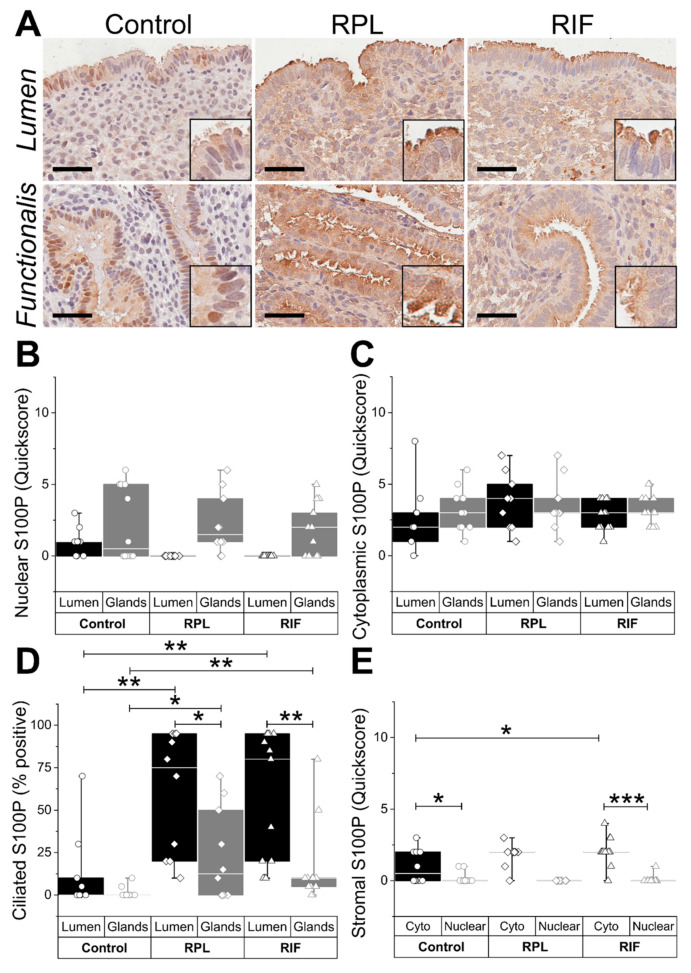
Endometrial S100P expression in control and recurrent reproductive failure. (**A**) Representative micrographs demonstrate S100P immunostaining in endometrial tissue from fertile (control), recurrent pregnancy loss (RPL), and recurrent implantation failure (RIF) patients. Scale bar 50 μm. Immunostaining scores for (**B**) nuclear, (**C**) cytoplasmic, and (**D**) ciliated S100P expression in the luminal and glandular epithelium. (**E**) Immunostaining scores for nuclear and cytoplasmic S100P expression in endometrial stroma * *p* < 0.05, ** *p* < 0.01, and *** *p* < 0.001.

**Figure 4 ijms-22-03835-f004:**
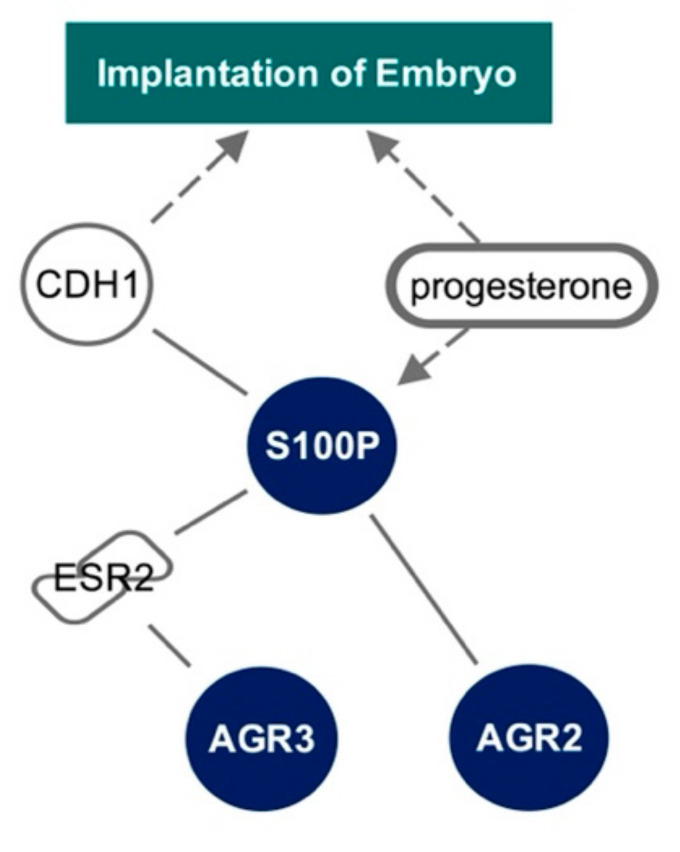
Pathway analysis to illustrate the relationship between S100P, AGR3, AGR2, and embryo implantation, using the Ingenuity Knowledge Base. Solid lines represent direct interactions. Dashed arrows represent indirect causation or expression.

**Figure 5 ijms-22-03835-f005:**
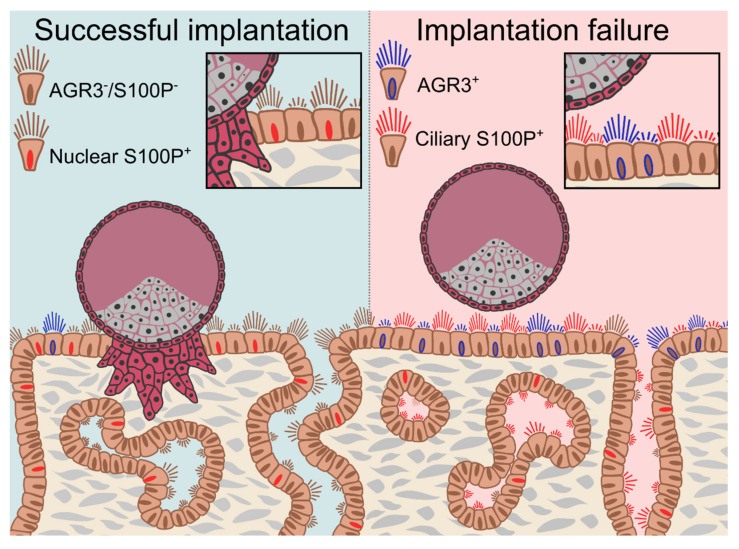
Differential AGR3 and S100P expression patterns are features of endometrial receptivity and may play a role in recurrent reproductive failure.

**Table 1 ijms-22-03835-t001:** Demographics, cohort 1.

Sample	Proliferative Phase (*n* = 10)	Secretory Phase (*n* = 10)	Statistical Significance
Age Mean (SD)	41.5 (8.7)	44.7 (4.5)	>0.05
Live Birth Median (range)	2 (0–4)	2 (0–3)	>0.05

Mann–Whitney.

**Table 2 ijms-22-03835-t002:** Demographics, cohort 2.

Sample	Control *n* = 30	RPL *n* = 10	RIF *n* = 13	Statistical Significance
Age Mean (SD)	41.2 (7.6)	39.1 (2.0)	35.7 (4.4)	<0.05
Live Birth Median (range)	3 (1–4)	0 (0)	0 (0)	<0.01
Miscarriages Median (range)	0 (0–1)	3 (3–5)	0 (0)	<0.01
Failed Embryo Transfers Median (range)	0 (0–0)	0 (0–6)	8 (6–12)	<0.01
Ectopic Median (range)	0 (0)	0 (0–1)	0 (0–2)	>0.05

Kruskal–Wallis.

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
