# Peer review of "Anterior Gradient Protein 3 and S100 Calcium-Binding Protein P Levels in Different Endometrial Epithelial Compartments May Play an Important Role in Recurrent Pregnancy Failure"

_ijms, 2021, doi:10.3390/ijms22083835_

Round 1

Reviewer 1 Report

The authors have conducted a study on AGR3 and and S100P levels in endometrial tissues and investigated their roles in RIF and RPL. 

  1. Line 95, first line in results section, should say "AGR3 levels in healthy human.." or should indicated protein levels somewhere in the sentence. 
  2. Image showing acetylated tubulin staining in lumen (Fig 1C) is not  lined up correctly with images showing AGR3 and merge staining in that figure (1C). It looks like ac-tubulin staining image is cropped different compared to the other two images. From ac-tubulin image it looks like top part of lumen is stained while in reality it is the bottom (As shown in merge). Authors have to fix this. 
  3. From the images, it also looks like total levels of S100P and maybe AGR3 are higher in RPL and RIF in addition to differences in protein localization? Can you check by western blotting? Also looks like ciliated cells is more in RPL and RIF groups. Is this true? Authors should comment on these observations if true.
  4. Thickness of biopsies need to be mentioned (number of microns etc).

Reviewer 2 Report

The paper of Tempest et al. describes differences in expression pattern of AGR3 and S100P proteins in luminal endometrial epithelium of healthy women and women with a history of recurrent implantation failure and recurrent miscarriage. Basing on immunohistochemical evaluations the authors conclude that higher expression and aberrant localization of these proteins in ciliated cells may contribute to the pathogenesis of recurrent miscarriage.

The study is well designed and original, and the paper is very well written. Although it appears to be descriptive and lacking a mechanistic approach it provides new important data on the pathogenesis of miscarriage. I have no specific comments.
